



# Air–sea carbon flux from high-temporal-resolution data of in situ CO₂ measurements in the southern North Sea

Steven Pint[1], Gert Everaert[1], Hannelore Theetaert[1], Michiel B. Vandegehuchte[1], Thanos Gkritzalis[1]

[1] Flanders Marine Institute, Wandelaarkaai 7, B-8400, Ostend, Belgium

*Correspondence to*: Steven Pint (steven.pint@vliz.be), Thanos Gkritzalis (thanos.gkritzalis@vliz.be)

**Abstract.** An important element to keep track of global change is the atmosphere–water exchange of carbon dioxide ($CO_2$) in the ocean as it provides insight in how much $CO_2$ is incorporated in the ocean (i.e. the ocean as a sink for $CO_2$) or emitted to the atmosphere (i.e. the ocean as a source). To date, only few high-resolution observation sets are available to quantify the spatiotemporal variability of air–sea $CO_2$ fluxes. In this study, we used observations of $pCO_2$ collected daily at the ICOS station

Thornton Buoy in the southern North Sea from February until December 2018 to calculate air–sea $CO_2$ fluxes. Our results show a seasonal variability of the air–sea carbon flux, with the sea being a carbon sink from February until June switching to a carbon source in July and August, before switching back to a sink until December. We calculated that the sink was largest in April (-0.95 ± 0.90 mmol C m$^{-2}$ d$^{-1}$), while in August, the source was at its maximum (0.08 ± 0.13 mmol C m$^{-2}$ d$^{-1}$). On an annual basis, we found a sink for atmospheric $CO_2$ of 130.19 ± 149.93 mmol C m$^{-2}$ y$^{-1}$. Apart from region- and basin-scale

estimates of the air–sea $CO_2$ flux, also local measurements are important to grasp local dynamics of the flux and its interactions with biogeochemical processes.

## 1 Introduction

Increased anthropogenic emissions of greenhouse gases (GHGs) lead to global warming (IPCC, 2019), and observing their balance is an important way to keep track of global change (Steinhoff et al., 2019). A key element in this balance is the air–

sea exchange of $CO_2$ in the ocean, as the oceans are responsible for the uptake of 25% of anthropogenic $CO_2$ emissions (Friedlingstein et al., 2019). The air–sea $CO_2$ flux provides insight in how much $CO_2$ is added to the marine environment from the atmosphere (i.e. the sea being a sink for atmospheric $CO_2$) or emitted by the marine environment to the atmosphere (i.e. the sea being a source). The North Atlantic Ocean is one of the major sinks with an uptake of 680 mmol C m$^{-2}$ y$^{-1}$ (Watson et al., 2009) in 2005, and between 800 and 4000 mmol C m$^{-2}$ y$^{-1}$ (Woolf et al., 2019) in 2010. Continental shelfs are regarded as

sinks of carbon with an average air–sea $CO_2$ rate of 1900 mmol C m$^{-2}$ y$^{-1}$ for the European continent (Borges et al., 2006). However, the Southern Bight of the North Sea (SBNS), i.e. a European shelf sea, is shown as a source area in Thomas et al. (2004). The latter is in contrast with other studies that suggest that the SBNS and the whole North Sea can be regarded as a sink for $CO_2$ (Borges and Frankignoulle, 2002; Laruelle et al., 2018; Schiettecatte et al., 2007). The southern part of the North Sea includes the Belgian Continental Shelf (BCS), which is a well-studied area in terms of air–sea surface dynamics and carbon





biogeochemical cycling (e.g. Borges et al., 2019; Gypens et al., 2011, 2004). In terms of air–sea carbon fluxes, the BCS shifted from being a carbon source in the 1950s to a carbon sink in the 1980s (Gypens et al., 2009), with more recent source-sink turnovers (Gypens et al. 2011). Changing seawater physical and biogeochemical characteristics in the BCS result in seasonal patterns of air-sea $CO_2$ flux (Gypens et al., 2004, 2011). The dynamic nature of the BCS in terms of annual $CO_2$ fluxes, which were often based on short term measurements and simulated values in past studies, highlight the necessity of high-resolution

robust $CO_2$ observations. Therefore, in the present study, we monitored the local dynamics of $CO_2$ flux using high-temporal-resolution data of both partial pressure of $CO_2$ in the sea ($pCO_{2, sea}$) and in the air ($pCO_{2, air}$). Our aims were to quantify the air–sea carbon flux, to identify what drives the seasonality of the flux in a specific year and to identify the annual source–sink dynamics in a specific location of the BCS.

## 2 Materials and methods

The North Sea has a surface of 670.000 km² (EEA, 2015) of which the Belgian Continental Shelf (BCS) occupies about 0.5% (or 3.454 km²; Belpaeme et al., 2011). The BCS is relatively shallow with water depths gradually increasing to 45m from the Southeast towards the Northwest (Van Lancker et al., 2015). Apart from extreme observations, surface water temperatures vary seasonally between 5°C and 20°C. The salinity is strongly influenced by the river plumes of the Scheldt, Rhine, Seine and Meuse (Lacroix et al., 2004) and varies between 29 to 35 PSU.

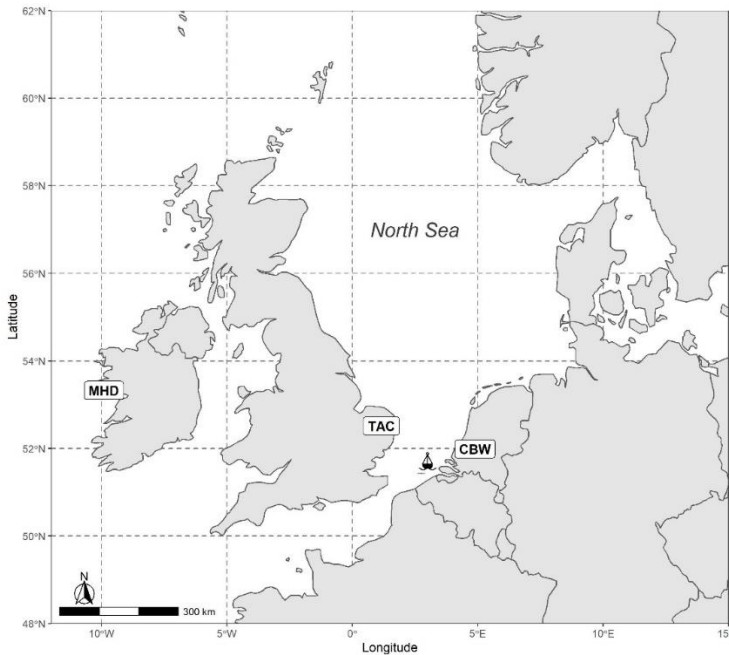


**Figure 1: Map of the North Sea with the location of the Thornton Buoy indicated by the buoy symbol. The locations of the ICOS atmospheric stations, i.e. Cabauw (CBW), Tacolneston (TAC) and the atmospheric station, i.e. Mace Head (MHD) are also presented.**



The Flanders Marine Institute is operating the Fixed Ocean Station "BE-FOS-VLIZ Thornton Buoy" in this area as part of the European research infrastructure "Integrated Carbon Observation System" (ICOS - ERIC, https://www.icos-cp.eu/). The station is equipped with commercial sensors to measure in-situ the sea surface $xCO_2$, atmospheric $xCO_2$, sea surface Temperature (SST), sea surface salinity (SSS) and the total gas pressure of $CO_2$. The BE-FOS-VLIZ Thornton Buoy is located approximately 30 km away from Zeebrugge in the area of the Thornton bank wind turbine farm (51.579N, 2.993E; Fig. 1). In

this study, we used observations from the year 2018. A schematic of the mooring and position of the sensors is depicted in Figure A1. The equipment details and data collection information are listed in Table 1.

**Table 1: Details of the in situ sensors on the BE-FOS-VLIZ Thornton Buoy and wind data collected from Meetnet Vlaamse Banken. The sensor (if available), sensor supplier, accuracy and sampling frequency are given for the following parameters; sea surface**
**salinity (SSS), sea surface temperature (SST), wind speed and partial pressure of both oceanic ($pCO_{2, sea}$) and atmospheric $CO_2$ ($pCO_{2, air}$).**

| Parameter | Sensor | Sensor supplier | Accuracy | Sampling frequency |
|---|---|---|---|---|
| SSS | SBE37-SMP-ODO | Seabird Scientific | $\pm$ 0.01 PSU | 15 min |
| SST | SBE37-SMP-ODO | Seabird Scientific | $\pm$ 0.001 ºC | 15 min |
| Wind speed | - | | $\pm$ 0.1 m s$^{-1}$ | 10 min |
| $pCO_{2, sea}$ | CO2-PRO ATM | Pro-Oceanus Systems Inc | $\pm$ 10 µatm | 4 h |
| $pCO_{2, air}$ | CO2-PRO ATM | Pro-Oceanus Systems Inc | $\pm$ 1 µatm | 4 h |

All the sensors (Table 1) are connected to a data logger (CR6, Campbell Scientific) which has a dedicated WiFi connection with a nearby wind turbine (approx. 250 m away). The wind turbine is connected to the managing company's servers (C-Power) via underground optic fibre cables and there is secure Ethernet connection with the VLIZ servers. This allows secure

and robust 2-way interactive communication with the buoy system and individual sensors, and provides the means to adapt sampling strategies of the sensors and identify issues very effectively.

The sensors used for this study (SBE37-SMP-ODO and CO2-PRO ATM) are calibrated by the manufacturers once per year. Additionally the $pCO_2$ measurements of the buoy were validated monthly against calculated $pCO_2$ values from measurements of Total Dissolved Inorganic Carbon (CT), Total Alkalinity (TA) and pH of manually collected samples. Water sampling

followed the SOP1 described in Dickson et al. (2007). TA, CT and pH were determined using the methodologies described in Dickson et al. (2007). For TA the method follows SOP3b of Dickson et al. (2007; commercially available system VINDTA 3s). The pH analysis and setup follows SOP6a (Dickson et al., 2007) using the Thermo Scientific Orion pH meter (STAR A211) and ROSS Sure Flow glass body pH electrode and we report pH at Total Scale at 25ºC as measurements are performed in a thermostatic environment (Grant Water Bath). Total Dissolved Inorganic Carbon (CT) is determined using the

commercially available Automated Infra Red Carbon Analyzer (AIRICA). For all methods, we use CRMs from Scripps Institute of Oceanography (UCSD). The uncertainties for each method are mentioned in Table 2. For the calculation of $pCO_{2, sea}$, we have used the R package 'seacarb' (Gattuso et al., 2020).



The calculated $pCO_{2, sea}$ values were used to calibrate the sensor data using a linear regression method (Fig. A2). The SST and SSS data of the buoy were validated against data obtained from RV Simon Stevin's CTD system (SBE3 & SBE4, respectively

for SST & SSS - Seabird Scientific) and underway Thermo-Salinograph sensor (SBE21 - Seabird Scientifics) when visiting the station and collecting samples. The $pCO_{2, air}$ measurements were evaluated against $xCO_2$ data from nearby ICOS atmospheric stations on land. For this comparison, we used the non-parametric Kruskal–Wallis rank sum test and the pairwise Wilcoxon Rank Sum test in the R package 'stats' (R Core Team, 2019).

**Table 2: Summary of analytical methods for analysis of spot samples**

| Parameter | System | System supplier | Accuracy |
|---|---|---|---|
| Total Dissolved Inorganic Carbon (CT) | AIRICA | Marianda | ± 2.5 µmol kg$^{-1}$ |
| Total Alkalinity (TA) | VINDTA 3s | Marianda | ± 2.0 µmol kg$^{-1}$ |
| pH (TS @ 25°C) | EMF method (Glass electrode + pH meter) | ThermoFisher | ± 0.0025 |

The air–sea $CO_2$ flux (F) is calculated (Eq. 1) according to the wind-driven turbulence diffusivity model of Nightingale et al. (2000) expressed in partial pressure:

$$F = k_{Nightingale} \cdot K_0 \cdot (pCO_{2,sea} - pCO_{2,air}) \tag{1}$$

where $k_{Nightingale}$ is the gas transfer velocity (length · time$^{-1}$, Eq. 2), $K_0$ is the solubility of $CO_2$ in seawater (mass · volume$^{-1}$ ·

pressure$^{-1}$) and $pCO_{2, sea}$ and $pCO_{2, air}$ are the partial pressure of $CO_2$. We calculated $pCO_2$ by multiplying $xCO_2$ measurements with the total gas pressure of $CO_2$ respectively in seawater or atmosphere. The solubility of $CO_2$ in seawater depends on the sea surface temperature (SST) and the sea surface salinity (SSS; Wanninkhof, 2014). The gas transfer velocity of Nightingale et al. (2000) uses wind speed measured at 10 m height ($U_{10m}$).

$$k_{Nightingale} = 0.222 \cdot U_{10m}^2 + 0.333 \cdot U_{10m} \tag{2}$$

Wind speed (10m above sea level) data were acquired from Meetnet Vlaamse Banken (MVB) for the Westhinder platform, Wandelaar platform and Scheur Wielingen platform, which are located approximately 20 km to 40 km more to the South and Southwest. Wind speed was measured every ten minutes. In the SST and SSS records, there are no data in September 2018 due to a malfunction in the buoy's SBE37-SMP-ODO sensor. To account for the lacking SSS data, we completed our times series with salinity data from by the RV Simon Stevin's CTD system in the same period (Flanders Marine Institute, 2019).

The SST data gaps were completed by data from a second water temperature sensor installed on an Aanderaa Seaguard multiparametric platform (Fig. A1b). Timestamps were used in order to combine data sets from various sensors and systems. All data were assessed for potential outliers. As in Salgado et al. (2016), outliers were defined as values lying outside the borders of the lower quartile minus three times the interquartile range ($Q_{25} – 3*IQR$) and the upper quartile plus three times the interquartile range ($Q_{75} + 3*IQR$). The daily mean air–sea $CO_2$ flux was calculated from 1891 time points. We took the

day–night cycle of the $CO_2$ flux into account by using daily means. Besides, we calculated monthly means and standard





deviation. Our data covers the period from February 2018 until December 2018. To quantify the annual $CO_2$ flux based on eleven months of data, we calculated a weighted mean for the winter, i.e. February and December, and the remaining nine months, respectively, using weights 0.25 and 0.75. We, then, extrapolated the weighted mean to a year. A summary of the input data is provided in Table 3. We investigated if the $CO_2$ flux calculated with $xCO_{2, air}$ from the Thornton Buoy was

different from the $CO_2$ fluxes based on $xCO_{2, air}$ measurements at nearby atmospheric stations (Sect 3.1). We compared them using the non-parametric Kruskal–Wallis rank sum test and the pairwise Wilcoxon Rank Sum test in the R package 'stats' (R Core Team, 2019).

**Table 3: Summary statistics of the variables, i.e. sea surface salinity (SSS), sea surface temperature (SST), wind speed and partial**
**pressure of both oceanic (pCO$_{2, sea}$) and atmospheric CO$_2$ (pCO$_{2, air}$) that were used to calculate air–sea CO$_2$ flux. In total, we used 1891 measurements of each variable in this study.**

| Parameter | Min | Max | Mean |
|---|---|---|---|
| SSS (PSU) | 32.1 | 34.7 | 33.4 |
| SST (°C) | 3.3 | 22.2 | 13.4 |
| Wind speed (m s$^{-1}$) | 0.3 | 17.9 | 6.8 |
| pCO$_{2, sea}$ (µatm) | 126.9 | 525.6 | 337.8 |
| pCO$_{2, air}$ (µatm) | 389.4 | 464.7 | 418.0 |

We adopted the method developed by Takahashi et al. (2002) to separate and assess the seasonal effects of biological processes and temperature on $pCO_2$ and $CO_2$ flux dynamics over an annual cycle. We applied eq. 3 and 4, where $T_{mean}$ is the mean annual temperature (13.4 °C) and $T_{obs}$ is the in situ temperature. The relative importance of the components effects is expressed by
the thermal–biological ratio (T/B) or the difference (T-B), where T is $pCO_{2, therm}$ and B is $pCO_{2, bio}$.

$$pCO_{2, bio} = pCO_2 \cdot \exp(0.0423 \cdot (T_{mean} - T_{obs})) \tag{3}$$

$$pCO_{2, therm} = pCO_{2, mean\ annual} \cdot \exp(0.0423 \cdot (T_{obs} - T_{mean})) \tag{4}$$

A T/B ratio between zero and one implies the dominance of biological processes over thermal effects (T-B < 0), whereas a T/B ratio larger than one implies that temperature effects are dominant (T-B > 0).

**3 Results and discussion**

**3.1 Environmental conditions**

In 2018, sea surface salinity (SSS) varied between 32.1 PSU and 34.7 PSU (Fig. 2a) with a mean value of 33.4 ± 0.58 PSU. The water temperature followed a seasonal pattern with high temperatures in summer time (max. 22.2 °C) and low water temperatures during the winter (min. 3.3 °C; Fig. 2b). No seasonal pattern was observed for the wind speed, i.e. the wind speed
is highly variable throughout the year. The lowest wind speed measured was 0.3 m s$^{-1}$ and the highest was 17.9 m s$^{-1}$ (Fig. 2c).





The pCO$_{2, air}$ fluctuated between 389.4 µatm and 464.7 µatm (Fig. 2d). The pCO$_{2, sea}$ data were validated against calculated values of pCO$_{2, sea}$ from DIC and pH values of manually collected samples. After that the pCO$_{2, sea}$ data were corrected with a linear regression method and validated against the manually collected (spot) samples. The pCO$_{2, sea}$ had a large range (126.9 µatm – 525.6 µatm), and reached its lowest value in May and highest value in August (Fig. 2d).

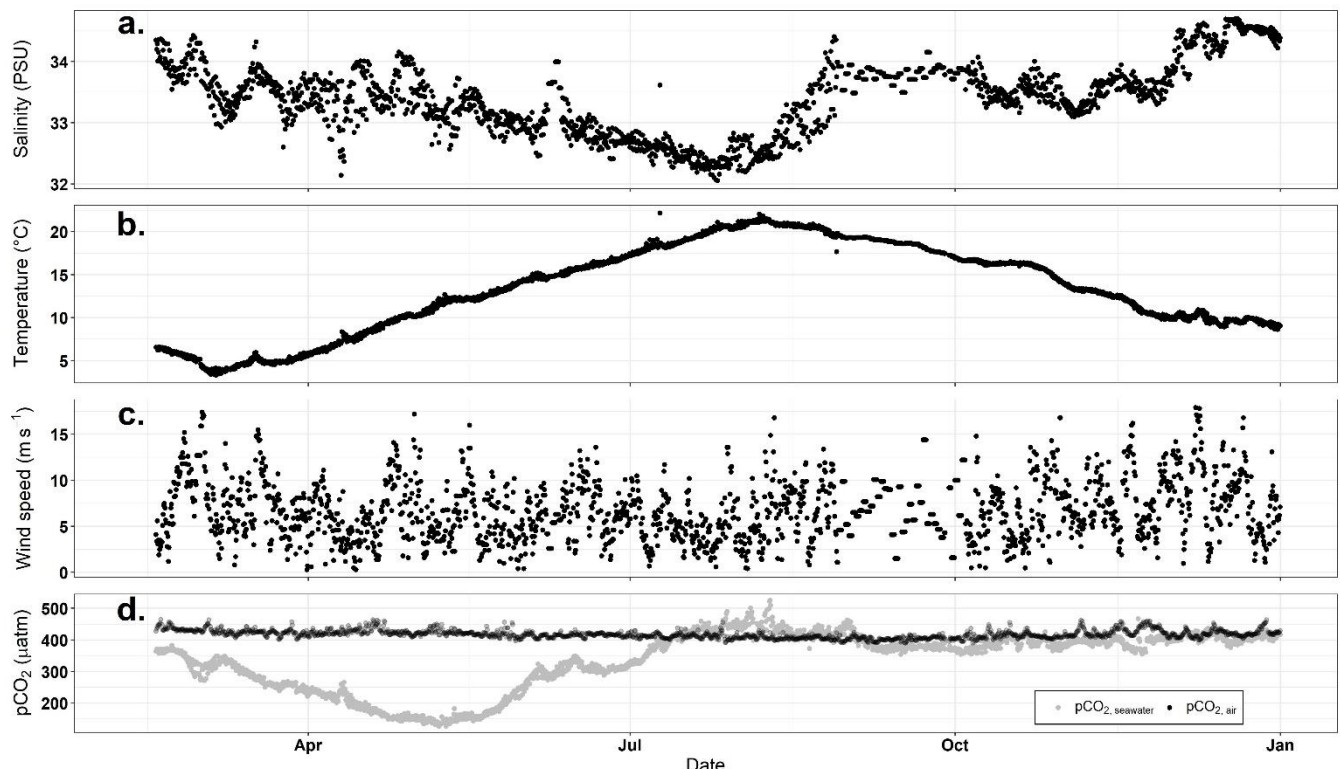


**Figure 2: All observations used in this study to calculate CO2 fluxes: (a) salinity, (b) temperature, (c) wind speed and (d) pCO₂ of seawater (black) and air (grey). The 1891 observations for each variable are given in chronological order from February 2018 to December 2018. The x–axis is the same for the different variables. All observations went through a quality control and correction to address and eliminate possible outliers from this study.**

These observed pCO$_{2, sea}$ concentrations corroborate with data found by Gypens et al. (2011) and Borges et al. (2006) for the English Channel (ECH) and the Southern Bight of the North Sea (SBNS). Borges et al. (2006) found that the spring bloom in early spring was followed by an increase in pCO$_{2, sea}$ in late spring–summer. Schiettecatte et al. (2007) observed that the SBNS was oversaturated in CO$_2$ during winter and strongly undersaturated in April–May. Schiettecatte et al. (2007) reported a minimum pCO$_{2, sea}$ value of 192.35 ± 35 µatm in the SBNS in April and a maximum of 455 ± 36 µatm in August. They

observed higher pCO$_{2, sea}$ values for the BCS, up to 900 µatm, but high values were measured close to the Scheldt plume (Schiettecatte et al., 2007). In the present research, we observed a seasonal trend of pCO$_{2, sea}$, which increased in the summer–early autumn and decreased in the winter–spring. We did not observe a strong seasonality in the pCO$_{2, air}$ record.

To evaluate our atmospheric xCO$_2$ data, we compared data from the BE-FOS-VLIZ Thornton Buoy with 2018 data from nearby (i.e. < 900 km) atmospheric stations (Fig. 3): i.e. two ICOS atmospheric stations Cabauw (207m above sea level;





Frumau et al., 2020) and Tacolneston (185m; O'Doherty et al., 2020) and one atmospheric station in Mace Head (24m; Delmotte et al., 2020). Usually, the $CO_2$ mole fraction data and products of these land-based atmospheric stations are used to calculate air–sea $CO_2$ fluxes, (e.g. Borges and Gypens, 2010). A basic comparison between the different data sets highlights the following. The minimum and maximum $CO_2$ mole fraction registered at the Thornton Buoy in 2018 was 389.4 ppm CO2 and 464.7 ppm $CO_2$. The atmospheric $CO_2$ mole fraction from sampling station Cabauw fluctuated between 394.0 ppm – 473.5

ppm $CO_2$, Tacolneston between 386.5 ppm – 455.1 ppm $CO_2$ and Mace Head between 394.2 ppm – 451.7 ppm $CO_2$. A similar trend was observed in the $xCO_{2, air}$ data of the Thornton Buoy as in the $xCO_{2, air}$ data of the other stations (Fig. 3). Our $xCO_{2, air}$ data is in range with the $xCO_{2, air}$ data of the land-based atmospheric stations (Fig. 3), which supports our use of local field observations. The use of local field observations of $xCO_{2, air}$ at sea provides useful information that complements the use of land-based stations because: 1) the sampling happens close to the water surface where the air–sea carbon exchange occurs,

and 2) the $xCO_{2, air}$ observations are more specific to the sampling location than land-based stations.

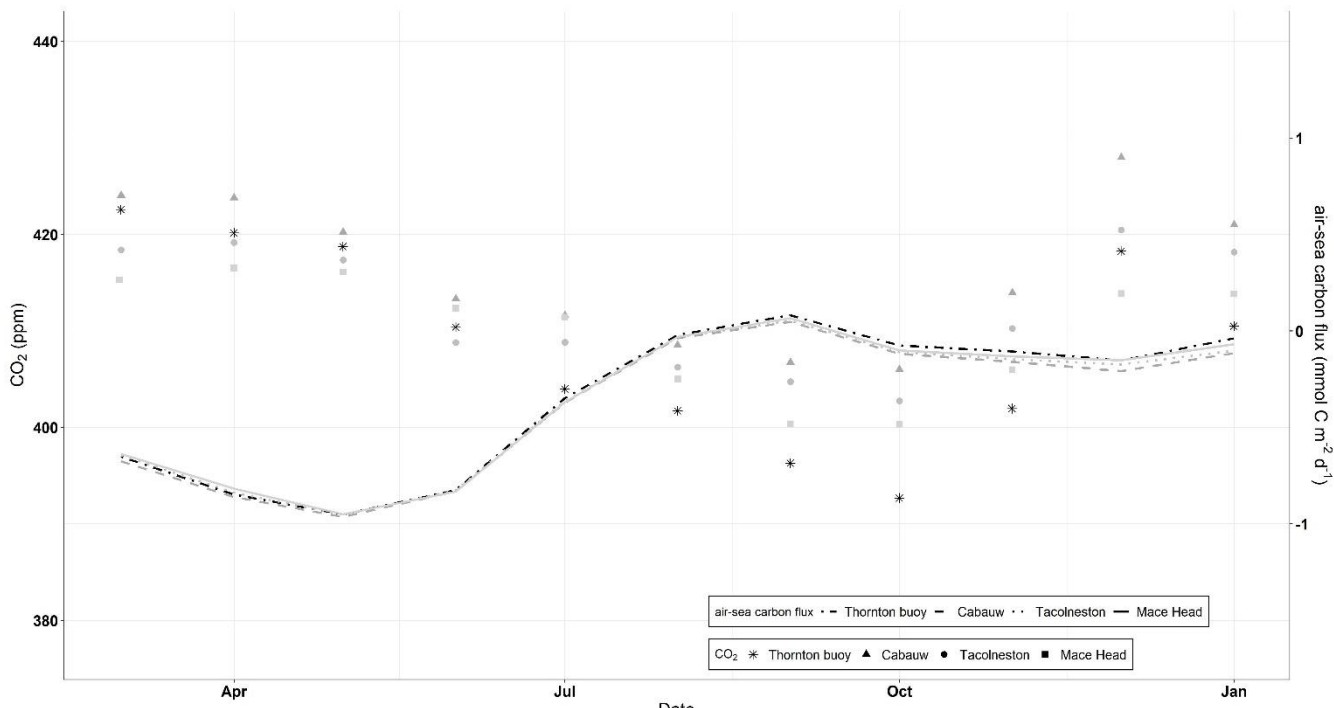

**Figure 3: Monthly mean of the $CO_{2, air}$ data and air–sea $CO_2$ flux based on $pCO_{2, air}$ data from the atmospheric measuring stations (greyshades); Cabauw, Tacolneston, Mace Head, and the Thornton Buoy (black) are given in a chronological order, i.e. from February 2018 to December 2018, for our study period. Error bars were not included to make the figure clearer. The left axis is used**

**for the $CO_2$ data and the right axis is used for the air–sea $CO_2$ flux.**

## 3.2 Air–sea CO2 flux

The air–sea $CO_2$ flux was estimated based on the salinity (Fig. 2a), temperature (Fig. 2b), wind speed (Fig. 2c), and pCO2 for seawater and atmosphere (Fig. 2d) time series at the Thornton Buoy in the BCS. We found that the wind speed had a large





impact on the magnitude of the $CO_2$ flux, i.e. higher wind speed increased the air–sea exchange of $CO_2$ in either way. The

daily means of the $CO_2$ flux varied between -2.99 mmol m$^{-2}$ d$^{-1}$ and 0.37 mmol m$^{-2}$ d$^{-1}$ (Fig. 4). We calculated monthly means

(-0.95 ± 0.90 mmol m$^{-2}$ d$^{-1}$ to 0.08 ± 0.13 mmol m$^{-2}$ d$^{-1}$) and distinguished a clear seasonal pattern (Fig. 4). We compared these

air–sea $CO_2$ fluxes with air–sea $CO_2$ fluxes calculated with $pCO_{2, air}$ of the atmospheric stations. Only the carbon flux using the

atmospheric $CO_2$ data of Cabauw differed from the carbon flux using $pCO_{2, air}$ of the Thornton Buoy (p = 0.031; Fig.3), showing

the importance of local atmospheric $pCO_2$ measurements. Overall, the air–sea $CO_2$ flux calculated with different $pCO_{2, air}$

sources, i.e. Thornton Buoy and atmospheric stations, followed a very similar seasonal trend (Fig. 3).

Coinciding with other studies in the SBNS, we noted a seasonal effect in the air–sea carbon flux (Borges et al., 2006; Borges

and Gypens, 2010; Gypens et al., 2011; Kitidis et al., 2019; Schiettecatte et al., 2007). The BCS at our location acted as a

carbon sink from February until June (-0.95 ± 0.90 to -0.34 ± 0.23 mmol C m$^{-2}$ d$^{-1}$; Fig. 4). The sink was the largest in April

with a monthly mean of 0.95 ± 0.90 mmol C m$^{-2}$ d$^{-1}$. The flux direction switches to a weak carbon source from mid-July and

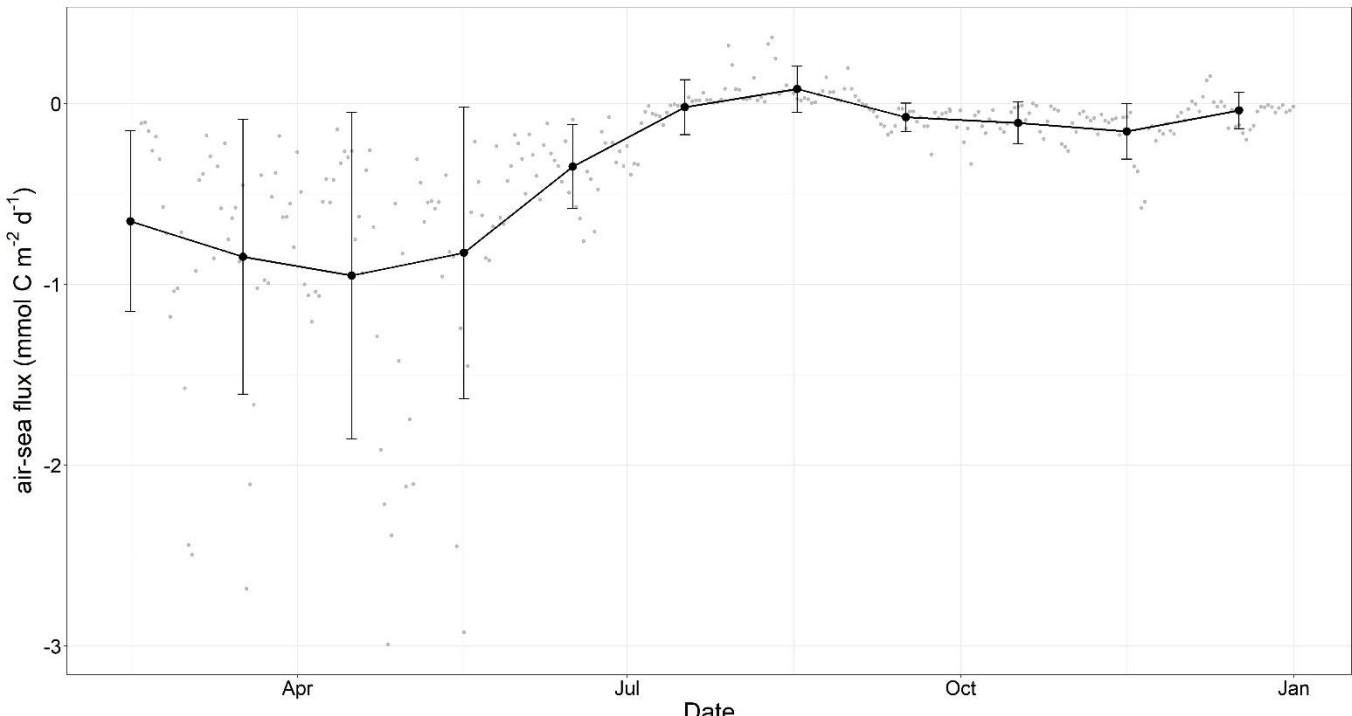


**Figure 4: The daily means of air–sea $CO_2$ flux measurements (grey) and monthly means (black) with for the study period of February to December 2018 is shown in chronological order. Error bars represent the standard deviation on the monthly mean. The x-axis marks indicate the start of the month. Note that the monthly mean is given at the middle of the month.**

until August with a monthly mean of 0.08 ± 0.13 mmol C m$^{-2}$ d$^{-1}$. However, our findings contradict with the other studies from

August onwards. We found that the BCS at our measuring station switched back again to a small sink from September until

December (-0.15 ± 0.15 to -0.04 ± 0.10 mmol C m$^{-2}$ d$^{-1}$; Fig. 4). We believe that the frequency and quality of our local

observations allowed us to identify the weak source in July and August, whereas it may have been unnoticed with different





observational capacity, e.g. sporadic sampling cruises. The Thornton Buoy ICOS setup allows for the collection of robust and high-frequency time series observations, , whereas sampling cruises can provide excellent spatial coverage however time

resolution can be sporadic (Borges and Frankignoulle, 2002; Schiettecatte et al., 2007). In that respect, it is possible that if samples and observations were obtained during a cruise in autumn over a relatively short period (days or weeks) when $CO_2$ was emitted, then the extrapolation of those observations could have led to the BCS being described as a source in autumn instead of a sink. We also need to acknowledge that environmental factors, e.g. temperature and biological activity can have significant effect on carbon fluxes (Gypens et al., 2011; Thomas et al., 2005, 2007; Wimart–Rousseau et al., 2020). Extreme

events, such as the heat wave in the summer of 2018, may have also contributed to some of the differences (e.g. increase in $CO_2$ concentrations) that we present in this study (Borges et al., 2019). Additionally, the solubility of $CO_2$ is lower in warmer water (Wiebe and Gaddy, 1940), reducing the uptake of atmospheric $CO_2$ (Yamamoto et al., 2018). Gypens et al. (2011) also simulated that the North Sea would change to a source for atmospheric $CO_2$ with warmer conditions (biological processes excluded). Other factors, such as wind and input of river plumes, are known to affect the air–sea carbon flux (Arndt et al.,

2011; Gypens et al., 2011; Laruelle et al., 2018; Nightingale et al., 2000; Thomas et al., 2005). High wind speed during the winter can amplify the $CO_2$-uptake in this season and so influence the yearly carbon exchange between the atmosphere and the sea (Kitidis et al., 2019). It is known that either temperature driven or biological processes are the dominant driving factor of the $pCO_{2,\,sea}$ (Schiettecatte et al., 2007; Thomas et al., 2005). In order to determine the main driver of the $pCO_{2,\,sea}$ dynamics, and as such to quantify the influence of temperature driven and biological processes on the observed $CO_2$ flux, we applied the

computational method of Takahashi et al. (2002).

We found that on an annual scale the biological activities dominated the $pCO_{2,\,sea}$ (T/B ratio = 0.69 and T-B = -113.32) and so $CO_2$ flux in the BCS. We also observed that the dominant factor changed by season. For the winter, i.e. February to March and October to December, we found that the thermal effect is dominant (T/B ratio = 1.24 and T-B = 42.28). However, in spring

and summer biological processes are dominant over the thermal effect (T/B ratio = 0.84 and T-B = -34.74). Our results correspond with the results of Schiettecatte et al. (2007), who found a T/B ratio of 0.74 (T-B = -70). This is, however, in contrast with the results reported by Thomas et al., (2005) who suggested that temperature rather than biological activity controlled the $pCO_{2,\,sea}$ dynamics seasonally. The data analysed in Thomas et al. (2005) were collected in four short term cruises and one cruise (i.e. in May) did not consider a $CO_2$ undersaturation (Schiettecatte et al., 2007). This $CO_2$ undersaturation

occurs in the declining phase of the phytoplankton bloom and is typically observed mid–April when the bloom is at its peak in the SBNS (Borges, 2003; Borges and Frankignoulle, 2002; Gypens et al., 2004). Based on our high-temporal-resolution measurements, we found that biological activities in BCS controlled the $pCO_{2,\,sea}$ and consequently the $CO_2$ flux (T/B ratio = 0.69). The high-temporal resolution is important to determine the seasonal variations in the $pCO_{2,\,sea}$, $CO_2$ flux and their underlying mechanisms. Linking high-temporal-resolution phytoplankton dynamics with our $pCO_2$ and $CO_2$ flux data

(Hilligsøe et al., 2011) may provide new insights in the $CO_2$ flux variation and its underlying drivers.





According to our data in our location, the air–sea $CO_2$ flux in 2018 was found to be mainly driven by biological processes, and we found that the BCS at our measuring station acted as a sink for atmospheric carbon on an annual scale (-130.19 ± 149.93 mmol C m$^{-2}$ y$^{-1}$). Our result is in line with other studies, who identified the SBNS as a $CO_2$ sink on an annual scale (Borges and Frankignoulle, 2002; Gypens et al., 2004; Kitidis et al., 2019; Schiettecatte et al., 2007; Fig. 5).

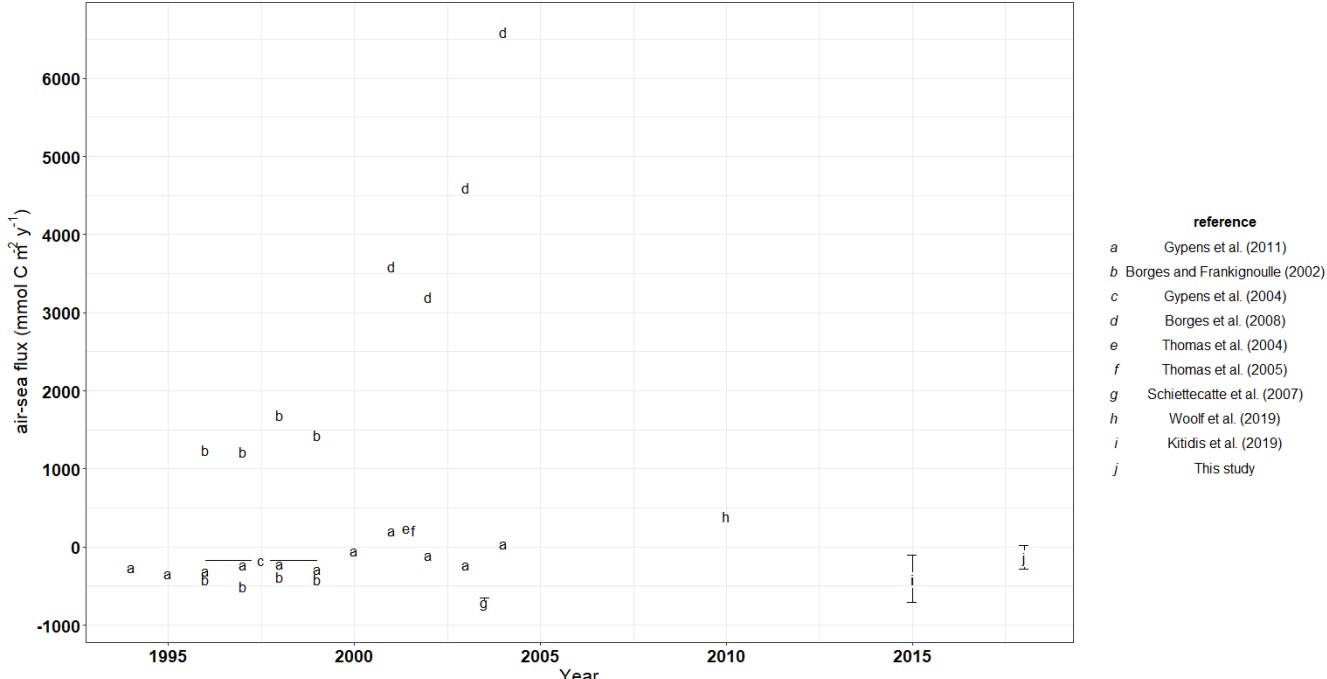

**Figure 5: The annual air–sea $CO_2$ flux from different studies (letter) with data from 1994 to 2018. Studies a, e, f, g, h and i provide an annual air–sea $CO_2$ flux for the SBNS, whereas studies b, c, d and j for the BCS. Please note that the high values (> 1000 mmol C m$^{-2}$ y$^{-1}$) of study b and d were located close (< 5 km) to the coast near Zeebrugge. Where possible, the standard deviation (or the standard error in case of study g) is shown by error bars. The horizontal line around a letter indicates that a mean was taken over the indicated period during that study.**

 Gypens et al. (2004, 2011) simulated annual $CO_2$ fluxes in range of our findings, e.g. -170 mmol C m$^{-2}$ y$^{-1}$ in 1996 – 1999 and -103 mmol C m$^{-2}$ y$^{-1}$ in 2002. However, the observed annual carbon sinks in other studies were twice (e.g. -300 mmol C m$^{-2}$ y$^{-1}$; Borges and Frankignoulle, 2002), to four (e.g. -700 mmol C m$^{-2}$ y$^{-1}$; Schiettecatte et al., 2007), to 20 times as large (e.g. - 2000 mmol C m$^{-2}$ y$^{-1}$; Kitidis et al., 2019) than our quantifications. Indeed, previous studies show a high inter-annual variability in $CO_2$ flux within the SBNS. Other studies (Borges et al., 2008; Thomas et al., 2004, 2005; Fig. 5) have observed that, in contrast to our study, the southern North Sea was a source of atmospheric $CO_2$ on an annual scale, e.g. 220 mmol C m$^{-2}$ y$^{-1}$ (Thomas et al., 2005). It should be noted that many of the $CO_2$ flux data of the southern North Sea are several years old, dating back from 2001–2002 (Thomas et al., 2005), 2003–2004 (Schiettecatte et al., 2007) and 2015 (Kitidis et al., 2019). These previous studies as well as our study show the high inter-annual variability in the BCS. The high inter-annual variability stresses the need to keep track of the air–sea $CO_2$ flux in a high dynamic area, such as the BCS. Having access to recent and high-temporal-resolution in situ data is important for robust coastal and ocean research but is also useful for policy makers, as



it could refine policy decisions. The carbon fluxes play a major role in the development of the ocean, i.e. ocean acidification (IPCC, 2019) and the global carbon cycle by absorbing anthropogenic carbon emissions (Friedlingstein et al., 2019). Our
findings are both in line, i.e. an annual sink for atmospheric carbon, and in contrast, i.e. an annual source for atmospheric carbon, with findings of others studies, demonstrating the high inter-annual variability.

The air–sea $CO_2$ flux does not only vary in time. It also varies in space. Having data on the spatial variability on a local scale, e.g. Thornton Buoy (this study) and Zeebrugge (Borges et al., 2008; Borges and Frankignoulle, 2002; Fig. 5), could be used
to assess the spatial variability within a larger area, such as continental shelf seas. Continental shelf seas showed an increase in absorbing atmospheric $CO_2$ and variability within the shelf, but also across different shelf systems (Landschützer et al., 2016; Laruelle et al., 2018). Though, it remains uncertain if the increase in atmospheric $CO_2$ absorption will continue (Legge et al., 2020). As global warming endures, seawater temperature will rise, consequently decreasing the solubility of $CO_2$ (Wiebe and Gaddy, 1940), reducing the uptake of atmospheric $CO_2$ (Yamamoto et al., 2018). In addition, global warming can affect
the $CO_2$ uptake indirectly by decreasing or stopping ocean circulation. Less ocean circulation will decrease the nutrient supply, weakening the biological processes and the $CO_2$ export (Yamamoto et al., 2018). The variability (Laruelle et al., 2018) and insufficient quantification (Legge et al., 2020) of air–sea $CO_2$ flux stresses the need for more and extensive in situ observations on local, such as in this study, and global scale (Bozec et al., 2006; Wimart–Rousseau et al., 2020; Woolf et al., 2019). High-temporal resolution of $CO_2$ flux monitoring is key to gain more knowledge about the inter-annual variability, its drivers and
the evolution of $CO_2$ flux. We also suggest extending the observations to investigate the spatial variability in the BCS.

## 4 Conclusion

We calculated monthly mean air–sea carbon flux at a station in the BCS using high-temporal-resolution data, i.e. daily measured values of $pCO_{2,\ sea}$ and $pCO_{2,\ air}$. By doing so, we revealed a large range of the variability in the air–sea carbon flux ($-2.99$ mmol m$^{-2}$ d$^{-1}$ and $0.37$ mmol m$^{-2}$ d$^{-1}$). The air–sea carbon flux displayed a seasonal pattern, with a sink in the winter–
spring months, a source in the summer and a small sink in autumn. We measured a carbon sink for atmospheric $CO_2$ in 2018 with an estimated uptake of $130.19 \pm 149.93$ mmol C m$^{-2}$ y$^{-1}$. We advocate for long-term sustained observations, that will allow to improve the quantification of coastal air–sea $CO_2$ flux and constrain the associated variations and drivers.



## Appendix A

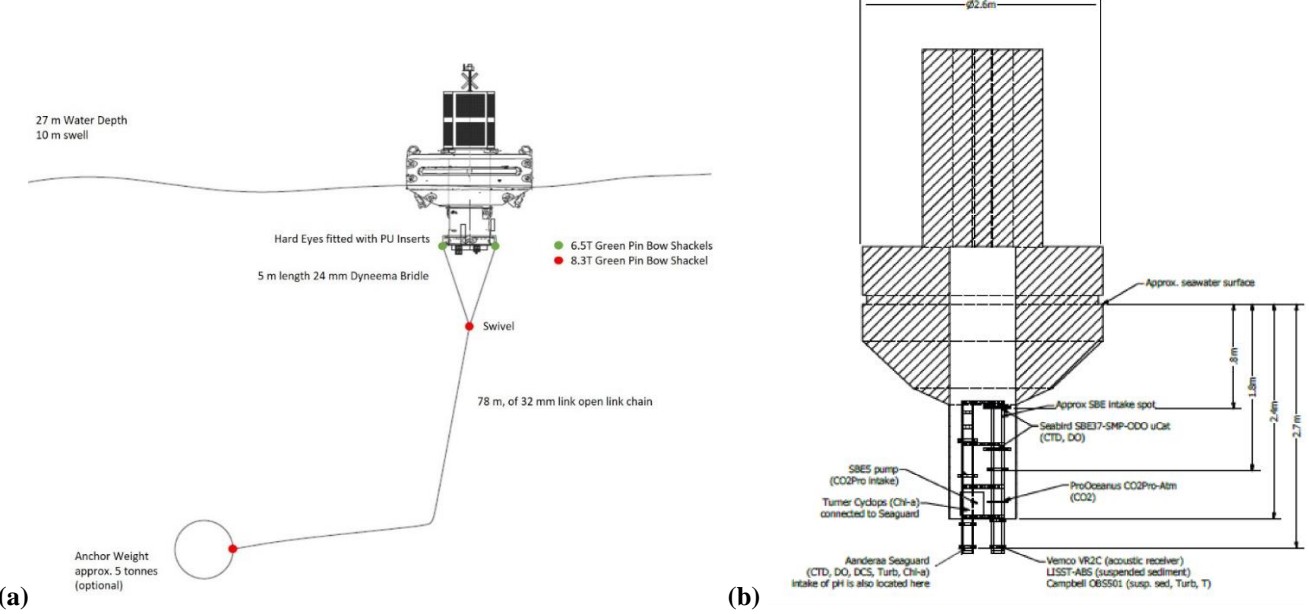

**(a)**                                                     **(b)**

**Figure A1: (a) Mooring details and (b) position of the sensors in Thornton Buoy. The sensors used in this study are provided in Table 1.**

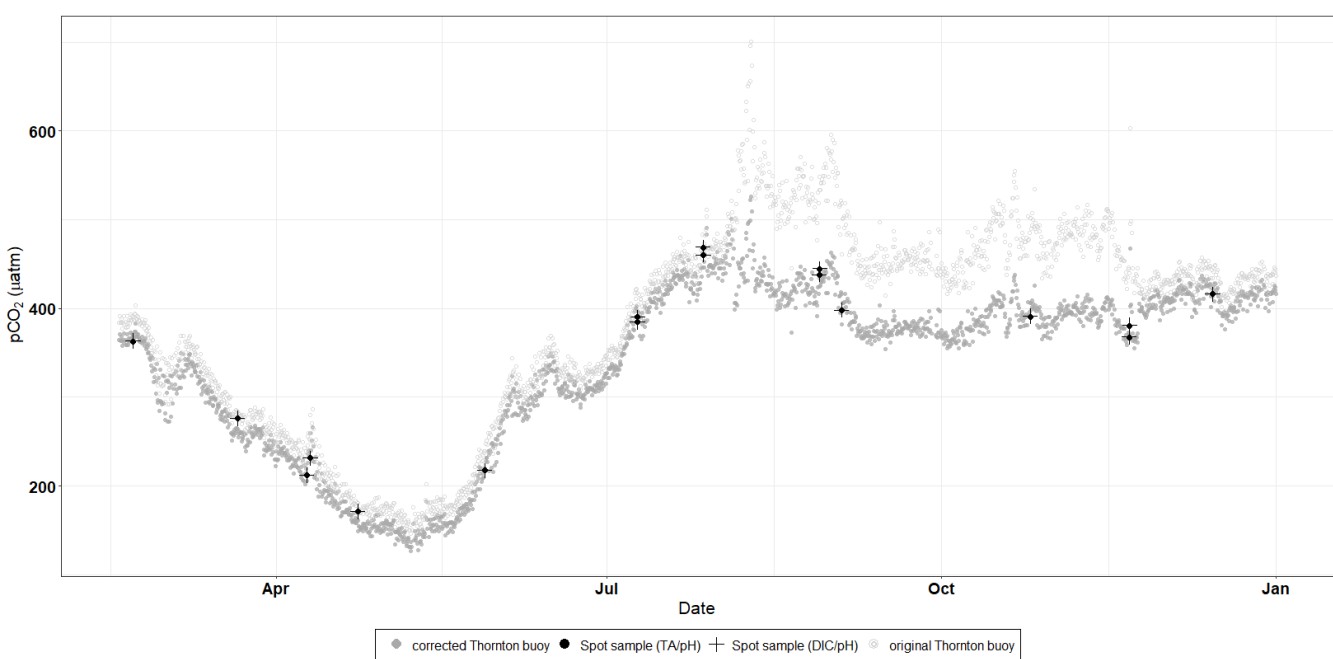

**Figure A2: The correction of the Thornton Buoy pCO$_{2, sea}$ measurements using a simple linear regression between the buoy's measurements and the spot samples, i.e. TA/pH and DIC/pH.**





## Data availability

Data is freely available from the corresponding authors (steven.pint@vliz.be, thanos.gkritzalis@vliz.be) upon request. The data will be available through the ICOS Carbon Portal soon. When available the authors will provide additional information.

## Author contributions

HT and TG performed the observations and quality control. SP processed the data and analysed the results in consultation with TG. SP wrote the manuscript in conjunction with TG, GE and MV. All authors discussed the results and gave their inputs to the manuscript. The VLIZ ICOS work is supported by Research Foundation Flanders (FWO) contract I001019N.

## Competing interests

The authors declare that they have no conflict of interest.

## Acknowledgements

We would like to thank the crew of the RV Simon Stevin (ICOS Station BE-SOOP-Simon Stevin) and the crew of DAB Vloot Zeetijger for their help on our monthly campaigns and deployment of the buoy. Many thanks also to ICOS OTC, which during the labelling procedure helped a lot to improve the station setup. Many thanks as well to Olivier Laurent (ICOS ATC) and Ute Karstens (ICOS CP) on their insights on our in situ atmospheric data, which we used on our discussions.

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
