# Peer review of "Air–sea carbon flux from high-temporal-resolution data of in situ CO2 measurements in the southern North Sea"

_Biogeosciences, 2020_

## Referee Comment (RC1) · Anonymous Referee #1 · 17 Dec 2020

In principle, I welcome new data. On the other hand, my immediate reaction after reading the manuscript is "so what". I fail to read anything that is not already known for the North Sea. It is the responsibility of the authors to clearly demonstrate what is new, other than just presenting a new set of data from a single buoy. My second concern is that there is a huge correction of the original data. The assumptions made in these corrections must be clearly stated. Another thing is that the study is for a single buoy in a well-studied sea. There should be some comparison with other shelves to make the study more significant in the context of global change.

---

## Referee Comment (RC2) · Anonymous Referee #2 · 4 Jan 2021

The manuscript by Pint et al. describes a 10 month long time-series of CO2 observations from a coastal mooring in the southern North Sea. From the 10 month time-series the authors infer an annual cycle of CO2 fluxes.

The coastal ocean is highly variable over space and time, and shortage of data with sufficient temporal and spatial resolution is needed to understand the processes that are at play. In that sense this data set is potentially important contribution complementing a number of earlier studies that are cited in this manuscript. Simply the fact that this is a temporally well-resolved data set would justify its publication.

However, the study fails on clarity for a number of issues. Most important is probably

the poorly explanation of the large (huge) corrections of the data from roughly July – December time-period. It seems to be a correction in the order of 100 uatm. Without that correction the flux would be more consistent with the large positive flux reported by Kitidis et al. (2019), for instance. I am not saying it is wrong, but it is suspicious, in particular since the correction seems to converge back to a very small correction for the later part of the time-series.

I don't think it is correct to infer an annual flux estimate and an annual cycle from a 10 month time-series (let alone from one with doubtful data for 4-5 months). What is the reason for only using these 10 months of data? Is the time-series aborted? If more data is available, I would strongly suggest to include at least one more year of data, if not also the 2020 data (if available). That would make the conclusions more convincing. Now I am not that convinced at all.

The oceanographic setting of the stations is poorly described in the text. Noting that the station is close to the Rhine estuary and the large difference in terms of CO2 fluxes one can expect from an estuary- relative to a shelf-station. What is the influence of river run-off on the observations, and how much of that driving the variability that is observed at the station, and how much can that drive inter-annual variability (different flow and pathways of the river water)?

Minor issues: Line 53: On the sensors. I would recommend more information on the sensors, calibration etc. Simply stating "commercial sensors" is not convincing although more information is available in Table 1. However, how can the authors claim an accuracy of +/- 10 uatm, and then do a correction of almost 100 uatm? I am sure this is the stated precision from the manufacturer, and evidently there are some issues with the instrument, at least for a part of the time-series.

Line 169: It is not surprising you found higher fluxes for higher wind-speed. . .. That is so obvious from the flux calculation the authors used.

Line 223: How did the author determine the flux of anthropogenic carbon? The measure pCO2 and have no way of teasing out the anthropogenic carbon flux from the natural.

Line 235: The authors refer to high inter-annual variability, so that makes even more of an argument to include more data in this study.

The data will have to be available at ICOS before this article can be considered for publication.

---

## Author Comment (AC1) · 15 Jan 2021

We appreciate and welcome the referee's comments and we will address them in a revised manuscript. In general, the referee mentions the lack of novelty, the correction applied to the original data, and the single buoy origin of the data as most important remarks.

To better highlight the novelty of the study in the revised version of the manuscript, we emphasize and highlight the strengths and weaknesses of the proposed setup. We are not aware of any other station or platform in the Southern Bight of the North Sea or nearby regional seas with similar setup. Despite few setbacks (mainly related to

hardware failures, e.g. failure of the batteries and solar panels) the class 1 labelled ocean station is operational and with capacity to continue to provide quality assured and quality controlled data within the ICOS network.

We acknowledge that the area was well studied, however we have not come up with datasets from recent years (i.e. 2015 onwards) and with similar sampling frequency. Especially in the past decade, the ocean abiotic conditions are changing at unprecedented rates.

We will include in the revised manuscripts air-sea carbon flux information from different continental shelves and global estimates.

The reported correction of pCO2, sea measurements is based on a simple linear regression between in situ measurements and spot samples collected when the station was visited with our research vessel RV Simon Stevin. We have identified that the sensor values were closer to the spot samples from February 2018 until July 2018 and then there is a larger deviation from August 2018 until November 2018. This is because of increased biofouling after a prolonged deployment. Once this was identified and conditions allowed, we resolved this during our maintenance visits, by cleaning the sensor. The latter clearly improved the performance as can be seen in December 2018 (Fig. 1). In that respect, we have decided to use 2 linear regression periods. To make the corrections, we applied one regression curve for the period February 2018 – July 2018 and December 2018 (Fig. 2) and another regression curve for the period August 2018 – November 2018 (Fig. 3). We are also confident that the erroneous sensor values for the seawater CO2 are because of biofouling, pre and post deployment calibrations of the sensor's NDIR detector, performed by the manufacturer suggest minimum or no drift of the detector's signal. We will include the details of our corrections in the supplementary material of this paper.

**Biofouling removed**

pCO₂ (μatm) — the y-axis: $pCO_2$ (μatm)

Thornton Buoy sensor (CO2-PRO)   Spot samples

Date

**Fig. 1.** pCO2 form the Thornton Buoy sensor and spot samples. The removal of biofouling on the buoy's sensor (CO2-PRO) is indicated with the vertical dashed line.

$$y = 15 + 1 \cdot x, \ r^2 = 0.995$$

**Fig. 2.** Regression 1st period Feb-Jul 2018 and Dec 2018

$y = -76 + 1.4 \cdot x, \ r^2 = 0.672$

**Fig. 3.** Regression 2nd period Aug - Nov 2018

[Figure]

[Figure]

**Fig. 4.** Histogram of the absolute residuals for each period of correction

[Figure]

[Figure]

**Fig. 5.** The difference between the buoy's sensor and the spot samples. Negative values indicate a difference where the buoy's value is smaller than the spot sample value and vice versa

---

## Author Comment (AC2) · 15 Jan 2021

We welcome the reviewer's constructive comments that especially refer to the fact that the data are temporally well-resolved, and that only this justifies its publication. Next to few minor comments, the referee mainly mentions the poor oceanographic description of the study area, the lack of detailed description of the correction factors, and despite the high resolution a short period of data (10 months).

We acknowledge that the term annual that we are using is not consistent with the time series, therefore we will rephrase our terminology, e.g. 2018 flux. We need to inform the reviewer that the station is still active, yet there are some gaps in the data coverage

that are associated with hardware failures, e.g. failure of the batteries and solar panels, that we are constantly addressing.

The reported correction of pCO2, sea measurements is based on a simple linear regression between in situ measurements and spot samples collected when the station was visited with our research vessel RV Simon Stevin. We have identified that the sensor values were closer to the spot samples from February 2018 until July 2018 and then there is a larger deviation from August 2018 until November 2018. This is because of increased biofouling after a prolonged deployment. Once this was identified and conditions allowed, we resolved this during our maintenance visits, by cleaning the sensor. The latter clearly improved the performance as can be seen in December 2018 (Fig. 1). In that respect, we have decided to use 2 linear regression periods. To make the corrections, we applied one regression curve for the period February 2018 – July 2018 and December 2018 (Fig. 2) and another regression curve for the period August 2018 – November 2018 (Fig. 3). We are also confident that the erroneous sensor values for the seawater CO2 are because of biofouling, pre and post deployment calibrations of the sensor's NDIR detector, performed by the manufacturer suggest minimum or no drift of the detector's signal. We will include the details of our corrections in the supplementary material of this paper.

We acknowledge that the oceanographic setting of the stations and the influence of the Scheldt/Rhine estuary on our observations are insufficiently described in our manuscript and we have taken this issue up in the study area part of the manuscript. The new paragraph now reads as: "The anti-clockwise rotation in the North Sea brings seawater from the English Channel towards the north (Fig. 6). Run-off from the Seine is brought to our area by this water flow. The Seine influences the salinity in our area the most (Lacroix et al., 2007). The influence of the Scheldt estuary near our observation station is relatively small. The Scheldt's influence on the salinity at our observation station is little as shown in Fig. 7 (Borges et al., 2008) and by (Fig. 13 in) Debrye et al. (2010). Freshwater flowing out of the Scheldt estuary is transported mainly southwards along and close to the Belgian coastline, i.e. away from our observation station. In case of nutrients, the Seine also plays a major role, except for the Scheldt estuary and northern part of the BCS. However, the distribution of nutrients could not be extracted from the water flow as nutrients are not conserved over seasons as a result of biological activity (Lacroix et al., 2007)."

Minor comments: Line 53: We are confident that the erroneous sensor values for the seawater $CO_2$ are because of biofouling, pre and post deployment calibrations of the sensor's NDIR detector, performed by the manufacturer suggest minimum or no drift of the detector's signal. We will add a paragraph in materials and methods focused on the sensors, e.g. calibration.

Line 169: This will be omitted in the revised manuscript

Line 223: We are not using the term anthropogenic for our work as we are aware that we cannot do this with this type of measurements. This term is only used when referring to other work (e.g. Friedlingstein et al. 2019).

Line 235: In this study we use high-resolution robust $CO_2$ observations, which allowed to identify significant variability in surface water $pCO_2$; a variability that was also evident in the 2018 $CO_2$ flux. Thanks to our unique set-up we were able to do this including $pCO_2$, atm measured at sea close to the action zone (3 m). Unfortunately, we cannot provide a continued time-serie including 2019 and 2020 due to setbacks (mainly related to hardware failures, e.g. failure of the batteries and solar panels). The data collected in 2019 and 2020 is rather sporadic than continuously due to these setbacks.

As part of ICOS we are working with the ICOS Central Facilities (Ocean Thematic Centre and Carbon Portal) in order to make the data available in the ICOS Carbon Portal. This will be completed by the first quarter of 2021. We would also like to mention that data will be submitted to SOCAT v 2021.

Biofouling

removed

pCO$_2$ ($\mu$atm)

600

400

200

0

- Thornton Buoy sensor (CO2-PRO)   ○ Spot samples

Apr                    Jul                    Oct

Date

**Fig. 1.** pCO2 form the Thornton Buoy sensor and spot samples. The removal of biofouling on the buoy's sensor (CO2-PRO) is indicated with the vertical dashed line.

$$y = 15 + 1 \cdot x, \ r^2 = 0.995$$

**Fig. 2.** Regression 1st period Feb-Jul 2018 and Dec 2018

Y-axis: Spot samples pC O$_2$ (μatm), ranging from 200 to 600

X-axis: Buoy sensor samples pCO$_2$ (μatm), ranging from 200 to 400

$$y = -76 + 1.4 \cdot x, \; r^2 = 0.672$$

**Fig. 3.** Regression 2nd period Aug - Nov 2018

[Figure]

[Figure]

**Fig. 4.** Histogram of the absolute residuals for each period of correction

[Figure]

[Figure]

**Fig. 5.** The difference between the buoy's sensor and the spot samples. Negative values indicate a difference where the buoy's value is smaller than the spot sample value and vice versa

[Figure]

**Fig. 6.** The counterclockwise residual current in the North Sea (black) and the deep water flow from the Atlantic Ocean (gray) (Turrell 1992). Meer et al. (2016)

[Figure]

**Fig. 7.** Sea surface salinity isolines based on sampling from 1995 to 2004. The buoy symbol indicates the location of the Thornton buoy. The measuring station (Z) was used in Borges et al. (2008)